**Data Availability Statement:** Data cannot be shared publicly because it contains sensitive

# HEalth professionals Responding to MEn for Safety (HERMES): Mixed methods evaluation of a pilot sexual health intervention for gay, bisexual and other men who have sex with men experiencing domestic violence and abuse

Ana Maria Buller[1]*, Giulia Ferrari[2], Alexandra Bleile[3], Gene S. Feder[4], Petra J. Brzank[5], Loraine J. Bacchus[1]*

1 London School of Hygiene & Tropical Medicine, Department of Global Health and Development, London, United Kingdom, 2 London School of Economics and Political Science Centre for Peace, Women and Security, London, United Kingdom, 3 War Child Holland, Amsterdam, The Netherlands, 4 Bristol Medical School, Population Health, Science Institute, University of Bristol, Bristol, United Kingdom, 5 Nordhausen University of Applied Science, Nordhausen, Germany

* Ana.Buller@lshtm.ac.uk (AMB); Loraine.Bacchus@lshtm.ac.uk (LJB)

## Abstract

### Background

Domestic violence and abuse (DVA) is a violation of human rights that damages the health and well-being of—gay, bisexual and other men who have sex with men (gbMSM). Sexual health services provide a unique opportunity to assess for DVA and provide support. This study explores the feasibility and acceptability of Healthcare Responding to Men for Safety (HERMES), a pilot intervention aimed to improve the identification and referral of gbMSM experiencing DVA in a London NHS Trust.

### Methods

The before and after mixed method evaluation of the intervention included semi-structured interviews with 21 sexual health practitioners, 20 matched pre-post questionnaires, and an audit of 533 patient records to assess identification and referral of gbMSM experiencing DVA.

### Results

HERMES increased practitioners' self-reported preparedness and confidence in enquiring, identifying and responding to gbMSM experiencing DVA. HERMES increased staff awareness of DVA among these patients, which led to higher identification practices in their work. There was a significant increase in the identification and reporting practices of trained staff (0% to 30%), with 6 (5%) DVA cases identified. However, as far as we could determine, none of these patients contacted the support agency.

patient information that might compromise participants privacy. Although data might not directly identify our participants in combination it may become identifying in particular given the small sample size. Furthermore, we did not obtain consent from participants to share data outside the research team and Qualitative and quantitative data are available from the authors for researchers who meet the criteria for access to confidential data. Requests for access to this data will be assessed by the University of Bristol Data Access Committee and the PIs (requests should be made using the following link: https://url.uk.m.mimecastprotect. com/s/4ylsCD8vnC0DJmAH5r1qR?domain=forms. office.com)

**Funding:** This article presents independent research commissioned by the National Institute for Health Research (NIHR) (https://www.nihr.ac. uk) under its Programme Grants for Applied Research scheme (RP-PG-0108-10084) awarded to GF. The views expressed in this publication are those of the author(s) and not necessarily those of the NHS, the NIHR, or the Department of Health. The funder had no role in study design, data collection and analysis, decision to publish, or preparation of the manuscript.

**Competing interests:** The authors have declared that no competing interests exist.

## Conclusions

HERMES proved successful in raising staff awareness, provided tools that increased identification and a referral pathway to an external specialist DVA service for the LGBT community. However, the poor uptake of the referral service indicates a need for further exploration of the help-seeking behaviour of gbMSM experiencing DVA and whether they would prefer to receive support within a sexual health service. Reinforcement training and clinical supervision is needed to sustain positive changes in practice over time and address potential challenges posed by staff turnover. Initial training should be conducted through face-to-face sessions with a combination of in-person and e-learning materials and followed by in-person and online reinforcement activities.

## Introduction

Domestic violence and abuse (DVA) is a violation of human rights that significantly harms the health and well-being of—gay, bisexual and other men who have sex with men (gbMSM) [1–5]. The negative health consequences for gbMSM affected by DVA include depression, anxiety symptoms, and risky sexual behaviours, such as unprotected oral or anal sex and HIV transmission [4, 6–11]. A systematic review and meta-analysis conducted by Buller and colleagues in 2013 [8] reported a pooled lifetime prevalence of intimate partner violence (IPV) among gbMSM at 48% (95% CI: 31.23%-64.99%), encompassing physical, psychological, and/or sexual violence. More recently, a meta-analysis by Liu and colleagues [12] found a pooled prevalence of 33% (95% CI: 28%-39%) for IPV victimization, including all types of IPV and various recall periods. Liu and colleagues also reported a pooled prevalence of IPV perpetration among gbMSM at 29% (95% CI: 17%-40%) [12].

Despite seeking healthcare for DVA-related symptoms or injuries, individuals impacted by DVA often go unrecognised by clinicians, resulting in suboptimal care [13]. Considering the link between gbMSM DVA and sexual health risk behaviours, including HIV, [14–16] as well as depression/anxiety symptoms [3, 8, 17], sexual and mental health services have been identified as crucial entry points for DVA interventions within the healthcare system for this particular population. Nonetheless, healthcare practitioners, including those in sexual health and domestic violence prevention and response, often lack training on how to identify and respond to same-sex DVA [18–21]. A UK study revealed that sexual health clinics follow protocols for sexual assault, but fail to address sexual violence in other contexts, such as intimate partner relationships [21]. While DVA advocacy interventions by trained primary healthcare providers and specialist domestic violence organisations can be effective and cost-effective [22] in improving mental health outcomes and reducing violence for women [23, 24], there is limited evidence on suitable interventions for the gbMSM exposed to DVA.

In 2014, the National Institute for Health and Care Excellence (NICE) updated its guidelines entitled Domestic Violence and Abuse: Multi-Agency Working, with particular emphasis on the important role of service providers in recognising and responding to the specific needs of individuals who identify as gay, lesbian, bisexual and transgender [25]. The NICE guidelines recommend that healthcare staff, including those working in sexual health services, should be trained and ask patients whether they have experienced DVA as part of routine good clinical practice, *even when there are no indicators of such violence*. Furthermore, that staff should have access to information about services and formal referral pathways in place [25, p.15]. There

has been a growing body of evidence over the last two decades exploring how healthcare services can best respond to DVA experienced by women in heterosexual relationships. However, there has been a notable lack of research on healthcare interventions that focus on the needs of gbMSM living with DVA since the study reported in this paper, which was conducted between 2009 and 2012. This has resulted in the experiences of gbMSM being overlooked in current global discussions about how to strengthen the healthcare response to patients who experience DVA. The extent to which the 2014 NICE guidelines, as they pertain to gbMSM, has been implemented in UK healthcare settings is unknown. Consequently, there remains a critical gap in the current evidence base and further research is urgently needed to address this disparity.

This study aimed to assess the acceptability and feasibility of a multi-faceted pilot educational and support intervention for sexual health practitioners in a LGBT sexual health clinic in London. The intervention's objectives were to improve the identification of gbMSM experiencing and/or perpetrating DVA and referral to support. The term gbMSM is used to refer to male patients whose sexual orientation was self-reported as gay or bisexual [26].

## Materials and methods

### HERMES intervention

This study was part of the UK PROVIDE (Programme of Research on Violence in Diverse domestic Environments) research programme which The gbMSM workstream aimed to examine the prevalence and associated health outcomes of DVA in gbMSM attending sexual health clinics, and develop and pilot test an educational intervention, Health Professionals Responding to Men for Safety (HERMES), for health practitioners and care pathway for men [27]. The HERMES training for sexual health practitioners included content on: i) how to identify signs and symptoms consistent with DVA in gbMSM; ii) group discussion about what is different for, or specific to gbMSM experiences of DVA compared with heterosexual women and men who experience DVA, and what is different for, or specific to men who do not identify as gay or bisexual, but do have sex with men iii) national prevalence of DVA in gbMSM, health impacts and risk factors for DVA; iv) data on the prevalence of DVA among gbMSM attending the sexual health service based on a survey conducted in Phase 1 of the study, and their views on health practitioners asking gbMSM about DVA; v) practice in asking questions about DVA using a tailored designed flowchart (S1 Fig); vi) documentation of DVA in the patient clinic proforma including whether the patient was asked about DVA, their response and whether a referral was offered; vii) and a care pathway which included referral to the clinic health advisors who would act as the link to liaison with GALOP, a London-based anti-violence and abuse charity for lesbian, gay, bisexual and transgender (LGBT) people and finally viii) ongoing support for health care practitioners from a lead health advisor and a doctor. Thus, there was a two-step referral pathway which included a referral to the clinic health advisor in the first instance and a referral to GALOP if deemed necessary by the health advisor.

The intervention and training materials were developed by Respect UK (https://www.respect.uk.net/), the research team and two practitioners in the sexual health service (a health advisor and a doctor).The training was rooted in existing evidence and best practice for this population, including data on prevalence of DVA in gbMSM attending the sexual health service and their views on practitioners asking about DVA, from formative research conducted during Phase I of the study [27]. Posters and leaflets about DVA were placed in the waiting room areas.

## Setting

HERMES was implemented in a designated LGBT sexual health clinic at a London teaching hospital, which ran one evening a week. Sexual health practitioners at the clinic received two DVA training sessions three months apart and were offered two alternative dates for each session to accommodate different shifts and work patterns. Each training session lasted three hours and was attended by a total of 31 practitioners including eight doctors, 13 nurses and 10 health advisors. A trainer from Respect and two clinical co-trainers jointly delivered the training. The training team also included two members of the research team (AMB and LJB) who presented key findings from the study research, and a representative from GALOP the referral agency.

Before HERMES, the process at the After Five Clinic involved referring suspected cases of DVA to one of the Health Advisors, with these cases being manually recorded in the medical records without a specific place for DVA documentation, meaning it could be noted anywhere in the records. During HERMES, we maintained the basic process of referring suspected or disclosed DVA cases to a Health Advisor (as described in the previous section on training). The Health Advisor would provide a first-line response by discussing the patient's situation further and offering to make a referral to GALOP.

Only trained healthcare practitioners were required to implement HERMES in the specialist LGBTQ sexual health clinic. These trained practitioners were not required to implement HERMES in the generic sexual health clinics, which were primarily used by heterosexual patients, although we know that gbMSM would occasionally use those clinics if they were unable to get a timely appointment at the LGBTQ clinic. Staff were sometimes required to rotate between the generic and LGBTQ clinics, but those who did not receive training were not required to participate in HERMES.

## Data collection and analysis

We used a mixed method before and after evaluation design to explore the feasibility and acceptability of HERMES in relation to three research questions. Table 1 presents the research questions and associated data collection tools. Qualitative and quantitative data were analysed separately, and findings integrated at the reporting level [28].

Findings were drawn from the different data collection tools described in Table 1. Integrating different data sources facilitated a more nuanced analysis which allowed for data triangulation, with the semi-structured interviews helping to contextualise and explain the PIM questionnaire findings. In the following sections we provide methodological and procedural details for each data collection tool.

**PROVIDE Intervention Measure (PIM).** The PROVIDE Intervention Measure (PIM) was administered to sexual health practitioners before the start of the first training session (18th March 2012) and 3 months later during the reinforcement session (20th June 2012). The PIM questionnaire was initially developed to evaluate a training intervention for general practitioners in the UK [18]. The tool measured changes in practitioner attitude and practice with regards to identifying and responding to DVA in females (victims only), heterosexual males (victims and perpetrators), and males in same-sex relationships (victims and perpetrators). Furthermore, the questionnaire elicited information on (i) socio-demographic characteristics and previous training attended; (ii) preparedness to identify signs and symptoms of DVA, enquire about and respond to DVA, (iii) and practice issues including the number of DVA cases identified in the last three months, clinical presentations in which practitioners asked about DVA, DVA protocol use, and availability of DVA support services. The follow-up PIM questionnaire included an additional question on perceived usefulness of the HERMES

**Table 1. Research questions and data collection methods.**

| Research questions | Questionnaire (PIM) | Semi-structured interviews | Patient proforma | GALOP records |
|---|:---:|:---:|:---:|:---:|
| 1. Did HERMES increase practitioners' knowledge and awareness of DVA in gbMSM | x | x | | |
| 2. Did HERMES improve practitioners' preparedness to ask gbMSM about DVA and respond | x | x | | |
| 3. Did HERMES increase identification and referral of gbMSM patients | x | x | x | x |

PIM: Provider Intervention Measure

flowchart, an aid to support practitioners' decision making regarding DVA developed for the study (see S1 Fig).

Of 31 practitioners who attended training, 20 (65%) completed the pre- and post- PIM questionnaires. We used paired data on 20 participants to compare changes in attitudes and practice pre to post intervention. The analysis was conducted using Stata 14.1 [29]. Given the small sample size, non-normal distribution, and use of categorical variables/qin the question-naire, we report median scores pre and post intervention. We calculated bootstrapped medians of their differences with 95% confidence interval (CI) by re-sampling observations 50 times, with replacement. Wilcoxon signed-rank test was used to examine changes in attitudes and practice pre and post training. This test imposes no a-priori distributional hypotheses on data from paired samples and is appropriate as the number of observations in a pre-post-compari-son is not large. We did not distinguish between victims and perpetrators of DVA in the analy-sis relating to gbMSM males. As our analyses highlight, health practitioners have limited time during consultations to make this distinction (which should be the role of specialist DVA orga-nisations), and they encounter challenges in discerning victims from perpetrators due to the widespread occurrence of bidirectional violence in this population [30, 31]. Thus, all analyses compared changes in attitudes and behaviour in relation to three groups: females (victims only), heterosexual males, and males in same-sex relationships (victims and perpetrators). For males, we collapsed all questions that previously differentiated between victims and perpetra-tors of DVA.

**Semi-structured interviews.** Semi-structured interviews were conducted with a self-selecting sample of 21 of the 31 (67.7%) sexual health practitioners who attended the HERMES training (28th March and 20th June 2012), including the two clinical co-trainers. The interview explored practitioners' perceptions of the training programme and its impact on their practice, and support received. Interviews lasted between 2 to 3 hours and were conducted in a confi-dential clinic room by AMB and LJB. Interviews were recorded and transcribed verbatim and stored in NVivo 10 [32] for data management and coding. AMB and LJB initially read and annotated the same five transcripts using deductive and inductive approaches to develop a coding frame which was discussed and refined as new interviews were analysed. The remain-ing transcripts were coded by one researcher (AMB) applying the developed coding frame-work, but also allowing for new themes to emerge from the data.

**Audit of medical records pre-post intervention.** Questions regarding DVA were added to the male patient proforma (Fig 1) in the section containing questions on recreational drug use, number of sexual partners in the last three months and unprotected sex since the last test. The proformas were used for new patient visits and returning patients with new complaints. Medical records were audited between 15th September 2012 and 30th March 2013 by two trained clinic health advisors for a six-month period before and a six-month period

## History of Domestic Violence

Yes ☐                No ☐                Declined to answer ☐

Not asked as no symptoms ☐

Current ☐            Historical ☐

Physical ☐           Sexual ☐            Psychological ☐

Referral to health                        Yes ☐            No ☐
advisor?

External information/                      Yes ☐            No ☐
referral offered?

Did patient accept                        Yes ☐            No ☐
external referral?

**Fig 1. DVA questions in the male patient proforma.**

immediately after the intervention to examine changes in practice regarding asking about DVA and making referrals.

To avoid seasonal bias (i.e., the possibility that DVA disclosure is higher in some months) we chose the same time period before and after the intervention (April to September 2011 and April to September 2012). We developed a bespoke extraction tool and data entry manual to ensure consistency in data extraction. The extracted data included a unique ID code and the authors did not have access to information that could identify participants during or after data collection. Only the trained clinic health advisors accessed the medical records. AMB met regularly with the health advisors to conduct quality checks and ensure consistency in data extraction and interpretation.

The hospital IT department produced a listing of all the male patients aged 18 years and over attending the clinic during an 18-month period. In total, 307 patients attended during the pre-intervention period and 275 attended during the post-intervention period. We retrieved and extracted data from 553 (95%) of these 582 records (296 pre-intervention and 257 post-intervention). Health advisors used an Access database to store extracted data which was

subsequently analysed in Stata. We compared trends in DVA identification and referral before and after the intervention using means.

**Referral information from GALOP.**   To assess whether men who had disclosed DVA and were offered referral information about GALOP and had contacted the organisation, the research team liaised with GALOP staff to incorporate questions on source of referral in the history intake form, which all staff completed when they received a call. The questions made specific reference to the LGBT dedicated sexual health service at the London hospital. Referral from the specialist LGBT clinic was monitored for six months following the intervention.

## Ethics

Written informed consent was obtained from all participants after providing them with an information sheet and the opportunity to ask questions about the study. The study received ethical approval from the National Research Ethics Services Committee (Southwest Central Bristol) (11/SW/0315), from the London School of Hygiene & Tropical Medicine Ethics Committee LSHTM (5758) and from the London-based Hospital where the research took place (10/H0106/22).

## Results

We begin our results with a description of the sample, followed by evidence that addresses the three research questions outlined in Table 1. In the last sub-section, we present evidence on the barriers and facilitators to the implementation of HERMES.

## Participant characteristics

Table 2 presents the socio-demographic and background characteristics of the 20 practitioners who completed the pre and post PIM. Table 3 presents the socio-demographic characteristics of 22 practitioners who participated in a semi-structured interview.

**Table 2.  Characteristics of practitioners who completed pre-post PIM.**

| Characteristics | N (%) |
|---|---|
| **[a]Sex** | |
| Male | 5 (26.3) |
| Female | 14 (73.7) |
| **Age** (mean) | 35.6 |
| Age range | 23–53 |
| **Practitioner role** | |
| Doctor | 5 (25) |
| Nurse | 8 (40) |
| Health Advisor | 5 (25) |
| Health assistant | 2 (10) |
| **Patient load** | |
| Average patient no. per week | 59 (SD = 27.46) |
| **Groups included in previous DVA training** | |
| Females (victims) | 16 (80) |
| Heterosexual males (victims/perpetrators) | 10 (50) |
| Males in same-sex relationships (victims/perpetrators) | 3 (15) |

[a] Missing data

**Table 3. Characteristics of practitioners who participated in interviews.**

| Characteristics | N (%) |
|---|---|
| Male | 6 |
| Female | 15 |
| Doctor | 6 |
| Nurse | 8 |
| Health Advisor | 7 |
| Completed pre and post PIM | 14 |
| Completed only pre PIM | 6 |
| Completed only post PIM | 0 |

N = 1 did not complete pre and post PIM

## Did HERMES increase practitioners' knowledge and awareness of DVA in gbMSM?

Practitioners consistently expressed high levels of satisfaction with the training program. The experiential aspects of learning, specifically the utilisation of case studies and role plays, emerged as the preferred methods among the participants. The incorporation of case studies allowed practitioners to immerse themselves in real-world scenarios, providing practical context to their learning.

Practitioners found the training to be particularly effective in raising awareness of domestic violence and abuse (DVA) in male patients within same-sex relationships. The training also proved instrumental in helping practitioners recognise potential signs of being in an abusive relationship.

Through the exploration of case studies and interactive role plays, they developed a deeper understanding of the subtle indicators and red flags that may indicate an abusive dynamic.

This enhanced awareness and enabled practitioners to be more proactive in identifying and addressing DVA among male patients in same-sex relationships.

*I thought it was excellent. Well, it was raising awareness and the group became more aware of statistics, how often things happen, the kinds of questions we can ask, the way to ask questions kind of very useful having little doorways into approaching a subject that is otherwise quite difficult for a lot of patients, um and just very useful all round to raise our awareness really of services out there and how to pick up on subtle signals (I9).* [Health Advisor, Male]

Furthermore, the training focused on equipping practitioners with effective questioning techniques to sensitively broach the topic of abuse. By providing practical guidance and opportunities for practice, practitioners gained confidence in their ability to initiate conversations about DVA, ensuring that patients felt safe and supported.

*I really enjoyed the training itself because I felt more confident in addressing domestic violence and I understood the dynamics within MSM relationships or within the MSM community and what can happen better. I felt better able and more confident in being able to address it.* [Doctor, Senior House Officer, Female]

According to the PIM questionnaire, 15% (3/20) of surveyed practitioners reported receiving previous DVA training which included information on men in same sex relationships in contrast with 80% (16/20) reporting training on DVA in women. Most clinicians (74%, 14/19)

felt more efficient at their job in terms of being able to identify and respond to DVA (a question that was only included in the post-PIM) after the training. A quarter (5/19) of clinicians felt as efficient as ever, and no one reported feeling less efficient after the training. Only one health care practitioner did not answer this question.

These results correspond to practitioners' accounts during the interviews. The training enhanced practitioners' understanding of DVA in gbMSM. They gained a deeper understanding of the magnitude of the issue, various forms of abuse, how to identify signs, symptoms and risk factors, and the associated health consequences. Acknowledging that DVA was a legitimate health issue to be addressed in the clinic was also an important outcome of the training.

*I think what was most valuable was first of all raising it as [. . .]an issue that was relevant to medicine. I think even sometimes clinicians can see that [DVA] as being outside their remit and it's kinda like well, here is a sexual health clinic, we don't want to deal with any other issues, we've got enough to deal with, with patients attending for the services that we routinely offer.* [Consultant, Doctor, Male]

The invisibility of DVA among gbMSM prior to HERMES was evident in practitioner narratives which revealed underlying heteronormative views.

*I kind of more associate it [DVA] with men having multiple partners or even sort of unprotected sex [. . .]. We have a tendency to think of this [being related to] sexual desire, behaviour, risk taking. While if it's females who keep having unprotected sex we tend to feel somebody is forcing her to have unprotected sex, multiple times* [Clinical Nurse Specialist, Female]

The invisibility of DVA among gbMSM was also reflected in clinic tools, such as the male patient proforma, which did not contain a question on DVA before the HERMES intervention compared with the female proforma which did include a question. Practitioners spoke of how the training provided *new insights* that challenged their preconceived notions about the nature of DVA among gbMSM.

*No. To be honest, I'd never really thought about domestic violence and male patients [as victims], [. . .] typically you don't really think about it especially here. Like our proformas, there's a question prompt on the female pro forma [for DVA] but there isn't really a prompt on the male [proforma] so it's like . . . in your mind, that kinda makes you think, 'oh it's something you ask women, not really ask men', so I hadn't really.* [Staff Nurse, Female]

With regards to awareness of referral services, after the intervention the PIM questionnaire found an increase in the number of practitioners who reported being aware of adequate support organisations to which they could refer heterosexual or gbMSM male patients affected by DVA. However, this increase was not statistically significant (p = 0.125 and p = 0.625 respectively). Qualitatively we found that having a staff member from GALOP present in the training was critical to improving practitioner awareness of LGBT specific services and facilitated a direct connection. This helped to increase practitioner confidence about making referrals for male victims *and* perpetrators. This was particularly important as some practitioners pointed out that the victim-perpetrator status was not always clear when dealing with gbMSM where bidirectional violence was an issue–*it's actually more complicated. . .they don't know who's done what* [Staff Nurse, Female]. Whilst this ambiguity was a cause for concern, it was addressed during training and practitioners were advised to refer to the experts at GALOP rather than try to discern by themselves.

*I can remember the scenarios where people have disclosed, because they've been a victim and yet a couple of them had described having physical fights with their partner, you know, where it was a two-way event where they were coming to blows with each other. I guess I'd feel a lot more confident now if someone did disclose they were a perpetrator, knowing there is a service I could refer to, because previously it's like,"oh what do you do", get them to go see the health advisor. The health advisor probably thinks,"well thanks, thanks for that referral, what did you want me to do about it?*" [Consultant, Doctor, Male]

## Did the HERMES improve practitioners' preparedness to ask gbMSM about DVA and respond?

With regards to changes in practitioners' preparedness (i.e. asking, responding, identifying and referring) to deal with DVA, a non-parametric Wilcoxon signed-rank test suggested increased self-reported ability in dealing with all patients that may be at risk of involvement in DVA episodes in the PIM survey (Table 4).

Whilst there was an increase in perceived preparedness across all patient groups, the largest improvement was reported for gbMSM patients (p-values ranging from 0.0002 to 0.0064 across the four components of preparedness, see Table 4). The smallest improvement was observed in the management of female patients, in particular we did not observe any improvement in practitioners' preparedness to ask for female patients about their experiences of victimisation (p = 0.1458). Although practitioners reported an increase in new identifications of 'current or past DVA in the 3 months prior to the follow-up interviews' compared to baseline, according to our standardised scale, this increase was not statistically significant for any of the patient groups.

Following the training, health care practitioners felt more prepared to initiate discussions about DVA and pose relevant questions. There was a noticeable change in the clinic culture, with some practitioners expressing a sense of permission to inquire about male patients' experiences of abuse.

**Table 4. Pre-post changes in sexual health practitioners' preparedness to deal with DVA.**

| | n | Signed rank test p value | Baseline median | Follow-up median | Median of the differences | 95% CI |
|---|---|---|---|---|---|---|
| **Asking female (victims)** | 19 | 0.1458 | 4 | 4 | 0 | [-0.4, 0.4] |
| **Responding to female (victims)** | 20 | 0.0038 | 4 | 4.5 | 1 | [0.0, 2.0] |
| **Identifying female (victims)** | 20 | 0.0004 | 3 | 4 | 1 | [0.7, 1.3] |
| **Referring female (victims)** | 20 | 0.0094 | 4 | 5 | 0 | [-1.0, 1.0] |
| **Asking heterosexual male** | 20 | 0.0030 | 5 | 6 | 1 | [0.3, 1.7] |
| **Responding to heterosexual male** | 20 | 0.0019 | 5 | 6 | 2 | [0.8, 3.2] |
| **Identifying heterosexual male** | 20 | 0.0018 | 4 | 6 | 2 | [1.4, 2.6] |
| **Referring heterosexual male** | 20 | 0.0096 | 4 | 6 | 2 | [0.6, 3.4] |
| **Asking gay and bisexual male** | 20 | 0.0002 | 5 | 7 | 2 | [1.1, 2.9] |
| **Responding to gay and bisexual male** | 20 | 0.0003 | 5 | 6.5 | 1.5 | [-0.1, 2.1] |
| **Identifying gay and bisexual male** | 20 | 0.0004 | 4.5 | 6.5 | 2 | [0.9, 3.1] |
| **Referring gay and bisexual male** | 20 | 0.0064 | 5 | 6 | 3 | [1.6, 4.4] |

Note: perpetration and victimisation were collapsed during analysis for heterosexual males and for gay and bisexual males

This table reports the number of observations; the non-parametric signed-rank sum test of differences in the distributions between baseline and follow-up measurements; baseline and follow-up medians; and the bootstrapped median of the differences with its 95% confidence interval (CI) i.e. calculated by repeatedly sampling observations a number of times, in this case 50

*P: [. . .] it was really, really useful for me and I suppose because I had that case a couple of weeks previous to the training where I was kind of face to face with someone in that situation, it made me feel a lot more confident about how I'd approach it, it made me feel like I had permission to sort of kind of ask people sort of you know* [Senior House Office, Doctor, Male]

Moreover, establishing a personal connection with the support agency, GALOP, contributed to a sense of assurance among practitioners, as they knew they had a reliable referral option for men in same sex relationships.

The training also prompted practitioners to reflect on their role in addressing DVA and its inherent limitations. They came to understand that their responsibility was not to solve the problem but provide support and appropriate referrals.

*I think the [GALOP trainer] was very, very useful with regards to just, there was no stupid question, kind of reassuring us that it's ok to ask questions, and I think when you hear another professional saying that it gives you more confidence to, know that we will never solve the problem [of DVA], but it gives you the confidence to at least try.* [Health Advisor, Male]

### Did HERMES increase identification and referral of gbMSM patients?

Self-reported identification of gbMSM experiencing DVA, according to PIM, improved post training, although this change was not statistically significant. Prior to the training, twelve out of nineteen health practitioners reported regularly asking any group of patients about DVA when this was not spontaneously disclosed. After the training, fourteen out of twenty reported doing so (p = 0.6250). According to PIM, there was no statistically significant change in asking heterosexual male (p = 0.8495) or gbMSM (p = 0.5589) patients with DVA-related symptoms about DVA exposure. However, we found strong evidence that health practitioners asked female patients with DVA-related symptoms more often after the training, compared to baseline (p = 0.0294).

In general, many practitioners expressed the view that enquiry for DVA should be done routinely rather than relying on specific symptoms that could be challenging to recall. They further emphasised the importance of extending the practice of enquiry for DVA to include gbMSM who seek services at the generic sexual health clinics.

*I think it would be easier for the clinicians [referring to asking all gbMSM], it's just a model we used in MOZAIC [women's DVA intervention in the sexual health service]. It's something we're much more used to using. For example, HIV testing, we say that you don't do risk assessments. . .we're supposed to be screening everybody for HIV.* [Consultant, Doctor, Female]

Certain practitioners held the view that selectively asking men about DVA who presented with specific symptoms (e.g., repeat sexually transmitted infections) could be perceived as a form of judgement by the patients themselves. Furthermore, they felt that selective enquiry could result in missed cases of DVA among asymptomatic patients. Whilst practitioners were trained in selective enquiry for DVA, many chose to ask all male patients attending the LGBT sexual health clinic.

*You have to be careful that the person didn't feel like you were making judgements about them, you know, so maybe it depends on what your view of a lot of sexual infections is. I suppose that's a bit of a grey area. For me, the best way to do it, is just to ask it straight out, ask everyone this question. I'm the one in the original [training] session who said "We don't ask*

*this question to asymptomatic patients". Just because you don't have any symptoms doesn't mean you might not be in a domestic violence relationship. It is a bit odd that we only ask it to people that are symptomatic. It's a really good screening process, a really good opportunity to ask people, where their partner wouldn't expect them to be asked. It's a shame to lose that chance to ask so many people.* [Staff Nurse, Female]

Practitioners revealed that they were paying more attention to subtle cues and identifying more cases of DVA among gbMSM since the HERMES training. These cues were not limited to health problems (e.g., mental or sexual health) but also behavioural factors that might be indicative of past or current abuse experiences–*the whole thing about going to parties, taking drugs and having group sex and it's not always protected. I remember thinking that it's actually really good [to ask about DVA] because it's something you wouldn't necessarily pick up on* [Staff Nurse, Female].

*I: So, you feel your practice has changed a little bit from the things in training . . .*

*P: Yeah, I think so . . .I was here on Tuesday [and] when I came out in the waiting room, I saw two guys sitting there, one guy was sitting there with a black eye. I didn't see that man as a patient, but the first thing that sprang to mind was,"If this is the patient that I'm calling out, that's gonna be what my line of enquiry will be". "Has this arisen out of a domestic violence situation"? Whereas previously I would have just thought, "Oh this person got a black eye; it's probably not why they're here".* [Consultant, Doctor, Male]

*When I'm now seeing MSM patients with that same level of anxiety, is it because, and they're in a relationship, are they worried because their partner is putting them at risk because their partner is having other partners or there's other sort of power discrepancies within that relationship? So, yeah, it's given me a few extra triggers to ask them about these things, apart from an obvious black eye or someone who, you know, bursts into tears because, you know, the partner is doing something.* [Consultant, Doctor, Male]

However, practitioners felt it was crucial to maintain a vigilant approach to identifying DVA among gbMSM and their discovery of abuse was inherently intuitive. They acknowledged the varied manifestations of DVA among men and the importance of being attentive to even the most subtle indicators.

*I just saw [a patient] this morning and it was only through discussion with him, because I wasn't even thinking of domestic violence, but it was something he said. I thought oh let me ask some more probing questions to see what comes up, and he was very open to talking about it. But it's waiting for that cue, but it's easy to miss that cue.* [Health Advisor, Female]

An audit on use of the clinic proforma, which included a question about DVA for the HERMES intervention, suggested a measurable intervention effect, with 28% (95% CI, 15% to 41%) of weekly visits recording the use of a proforma after the training. The audit of medical records found no documentation of DVA before the intervention. Post-intervention practitioners enquired about DVA in 126 visits out of 415 (30%), as opposed to 0% pre intervention. Of the six patients who disclosed DVA, five were classified as historical cases, but no information on recency was available for the sixth; three reported physical and one psychological abuse, while two did not disclose type of abuse. In terms of practitioners' referral practices, two patients were referred to a health advisor in the clinic and one was offered external support information about GALOP but did not accept it. No information on practitioners' referral practices was

recorded for the patients who had either been exposed to psychological abuse or did not disclose the type of violence they were exposed to. Practitioners suggested that asking about DVA could be better integrated into the history taking consultation by placing the DVA questions alongside those on history of mental health and mental health interventions, or linked to medications (e.g., antidepressants)–*I think it would be easier to say, have there been problems at home, has there been any domestic violence, for example. That would flow better than where it is at the moment.* [Health Advisor, Male].

In contrast, referral practices did not change from pre to post training. Although GALOP was the referral service to which practitioners were asked to refer to. However, 44% of the clinicians (8/18) stated that information leaflets or cards for GALOP were well displayed and accessible to patients. Of the remaining 10 clinicians, four reported either that leaflets were not easily accessible, though available; and six that they were unsure about the availability of GALOP information materials at the clinic.

Despite the increased ability to identify cases of DVA, there were no referrals made to GALOP in the six months following the intervention. This could have been due to the high rotation of practitioners within the sexual health service, which meant that those attending the training did not have the opportunity to do many shifts at the designated LGBT clinic. Additionally, notes made by the Health Advisors in the database show that, of the three patients who were identified as experiencing DVA, one declined referral to the HA. When talking with the health advisors in the clinic, it emerged that although identification was happening patients declined to be referred.

Finally, practitioners mentioned that even when they managed to identify patients experiencing DVA, they were often reluctant to take up the referral. One of the clinical co-trainers recounted a patient who disclosed DVA, but became apprehensive when it was suggested that they have a follow-up conversation and that a referral to GALOP was one option:

> *"[. . .] when something gets identified the patient gets a frightening [. . .] they realise they've been ousted or something, they go 'oh God they've picked it up, this is serious', [. . .] taking in leaflets is then very symbolic of a problem and puts a label on them as victim whatever, so it's a tricky one."* [Senior Health Advisor, Clinical Co-Trainer, Male]

## Barriers and facilitators to implementation

Despite practitioners' increased confidence after the HERMES training, it was acknowledged that some practitioners may feel anxious when faced with a patient disclosure of DVA. There was a fear of opening a potentially complex and emotionally challenging situation, and uncertainty about how to effectively handle a disclosure.

> *I imagine fear in clinicians that you know; you are opening a can of worms here, what happens if they do go 'yes'? [. . .] And if you're pressing the button, what's gonna come out in the room. How are we gonna hold it?* [Health Advisor, Male].

This apprehension seemed to be heightened in the context of the busy clinic with heavy workloads, especially during shifts after five o' clock, where time constraints and an overwhelming number of patients added pressure. The demanding nature of their work and the need to efficiently attend to patients' medical needs further contributed to worries about addressing DVA in a sensitive manner. Another barrier to the implementation of HERMES was that of dealing with perpetrators of DVA. Despite training, several practitioners expressed discomfort and a sense of unpreparedness in dealing with men who admitted to perpetrating

DVA in gbMSM relationships, in contrast to working with victims - *I'd feel very, probably uncomfortable if somebody said,"I'm a perpetrator".* [Staff Nurse, Female] With regards to visual aids to support practitioners, whilst many thought that the HERMES flowchart (S1 Fig) was useful, very few reported referring to it during their encounters with male patients and preferred to have a simplified version. It was felt that there was little time to read it during busy appointments and some practitioners did not perceive it as a prompt for remembering key symptoms because the list was too long and there was no mnemonic to aid memorisation of it. It was suggested that computer prompts might be more helpful in alerting practitioners to ask when patients presented with certain symptoms and triggering other actions–*we should have access to that information at our fingertips.* For the most part, they could not remember what was in the flowchart or where it was located in the clinic.

> *I*: *So you think it would be better if it was on a computer?*
>
> *P*: *Yeah I think so and once again, if we were using that in the electronic notes, it could just be an automatic link to that, you click yes on this box, suddenly that pops up and you can go down that path.* [Consultant, Doctor, Male]
>
> *It's not easily retainable [referring to the HERMES flowchart]. The prompting conditions have to be in your head, so you're asking people to commit this list to memory, eleven prompting conditions. That's my issue with it, it's too long and hard.* [Consultant, Doctor, Female]

Conversely, according to the PIM questionnaire, many practitioners reported that the flowchart was useful in prompting them to enquire about DVA, reminding them about the common symptoms associated with DVA and the referral pathway. In the post PIM questionnaire, nine out of 20 practitioners reported that the HERMES flowchart was fairly useful and referred to it occasionally, whilst three said that it was not useful. Eight practitioners did not respond to this question. The DVA box in the paper-based male patient proforma was seen as a more helpful prompt for asking. The general consensus among practitioners was that sexual health clinics were good entry points for gbMSM seeking help for experiences of DVA, because practitioners and patients were accustomed to discussing sensitive issues in the context of sexual relationships during history taking.

> *It's because it's a sexual health clinic. . .asking quite intimate questions about their sex life and unprotected sex. . . [but] it's important people [know] you're actually asking [about DVA] for a reason and for their well-being. I've never had anybody being defensive.* [Health Advisor, Male]
>
> *Being able to check all the things pertaining to drug use, domestic violence, infections, risk taking. . .I've definitely been more focussed because it's good to have a new proforma, it's a bit more concise and clear in my role which is ongoing support, constant therapy with patients.* [Senior Health Advisor & Clinical Co-Trainer, Male]

Practitioners reflected on possible ways to improve the experience of HERMES, mostly around further training and modality of delivery. They expressed a need for more practice in asking about DVA. Despite their regular engagement in conversations with patients about highly personal matters within the sexual health setting, there was something fundamentally difficult about the topic of DVA that created discomfort. They wanted to find ways of integrating enquiry for DVA into their language scripts in a way that felt natural - *It's totally a matter of practice and trying to find a way of seamlessly linking into the flow* [Health Advisor, Male].

*I think yeah, I think instinctively there is something inherently different about asking questions about domestic violence rather than sexual practice because obviously the majority of the time sexual practice is something the person wants to take part in, domestic violence usually isn't so I think obviously the content of the questions is different.* [Clinical Nurse Specialist, Male]

It was proposed that practitioners would benefit from further reinforcement training to enhance their skills in addressing DVA, along with opportunities to discuss actual cases they were dealing with–a kind of clinical supervision. It was felt that this approach would promote experiential learning and guidance on how to respond appropriately in relation to a range of patient experiences of abuse. While practitioners were encouraged to reach out to GALOP for support in dealing with cases, there was no formalised or systematic process for incorporating this within the sexual health clinic. Others suggested that discussion about DVA cases could be part of the regular clinical team meetings. Online resources were suggested as a way for practitioners to obtain support in the absence of clinical supervision. In the absence of a formalised process, it was suggested that future refresher training should include more space and time for role plays.

*Because as I'm talking to you now it just struck me that if I had an [online] site I could visit when I'm not certain about something and just see what comes up and suggestions how to deal with that. That would help, like in nursing, we have, we have a site that for consultation sexual health gives you tips on how to allay people's anxieties and if a question comes up, or you want to ask a question that is difficult how to phrase it so if there's something like that for, it's a video session that you can access and watch a set of like clinicians.* [Nurse, Female]

Ongoing training was also seen as essential due to the frequent rotation of staff in the LGBTQ clinic. With staff often working on a one-in-six-week basis, they had limited opportunities to practice and enhance their skills in asking gbMSM about DVA and responding.

## Discussion

Since we conducted HERMES, there has not been another study of an intervention based in sexual health services specifically targeting gbMSM who experience DVA [11]. As such, this study still makes an important contribution to current research on DVA experienced by individuals who identify as LGBTQ and the need to address the broader socio-structural context that shapes their lived experiences of seeking help.

The mixed method evaluation of HERMES demonstrated that it was feasible and acceptable to staff. The training increased sexual health practitioners' confidence in asking all patients groups, including gbMSM, about DVA and responding to disclosures, with the greatest impact on their preparedness to handle gbMSM cases. The smallest improvement was observed in the management of female patients, with no improvement in the practitioners' ability to ask about female experiences of victimisation. This lack of improvement can be explained by the fact that sexual health practitioners had previously received training in a separate pilot intervention for female patients (2005–2007) called MOZAIC Women's Wellbeing Project [33].

The findings from both the qualitative interviews and PIM survey revealed a consistent trend, indicating an improvement in practitioners' self-reported capabilities and preparedness to meet the needs of gbMSM experiencing DVA. However, both the PIM survey and the pre- and post-intervention audit of medical records showed none or only a marginal increase in identified cases of DVA among men including gbMSM. Conversely, the PIM survey showed that identification of female patients increased after HERMES. Identification requires a higher level of skills and training and HERMES may have indirectly strengthened the previous

training on women. In comparison, the introduction of new training DVA among gbMSM may not have been enough to increase identification. Furthermore, the marginal increase in case identification observed in the medical notes may be due to factors outside the investigators' control, such as the mixture of trained and untrained practitioners in the clinic. Additionally, some practitioners may have identified DVA but did not record it in the proforma, or patients might have requested that it not be documented in their medical notes. Finally, since DVA identification was not routinely recorded in the medical notes prior to HERMES, it was challenging to ascertain the real impact of the intervention at this stage. A follow-up wave of data collection at a later stage would have allowed the intervention to become embedded into the service and to enable a direct comparison of identification recorded on the newly introduced proforma.

Moreover, none of the men who disclosed DVA during the HERMES intervention contacted GALOP, the designated referral organisation. This could be attributed to a combination of factors, including an ineffective referral pathway, slow adoption of the new male patient proforma, lack of trust in the healthcare response, and the possibility that men may not have been ready to seek help. Liang's [34] theoretical framework for understanding help seeking processes among survivors of domestic violence, proposes three non-linear stages: recognising there is a problem, deciding whether to do something about it and selecting a source of support. Thinking processes within each stage are shaped by individual, interpersonal and socio-cultural factors. Donovan & Barnes [35] argue that social factors related to cis-gendered heteronormativity, hetero-sexism and LGBTQ invisibility add an additional set of barriers to LGBTQ help-seeking. This finding is consistent with results from the formative phase of our study where 64% of gbMSM who reported experiencing at least one abusive behaviour on the Health and Relationship Survey, said that they had *never been in a domestically violent or abusive relationship* [27, p.7]. As discussed elsewhere, men in same sex relationships find it difficult to label their experiences as abuse because they do not recognise themselves in the 'public story' in which DVA is constructed as a problem in heterosexual relationships [36, 37]. The response of services also poses a barrier to gbMSM help seeking and includes homophobic attitudes among staff, misconceptions about the bidirectional nature of DVA between men in same sex relationships making it difficult to ascertain who is the perpetrator, or the belief that it is easier for gay men to leave an abusive relationship as they tend to move frequently from one partner to another [38, 39].

Our research findings suggest that certain barriers to integrating DVA response in sexual health services for gbMSM mirror those found in healthcare interventions targeting heterosexual women. These barriers include limited time, discomfort in asking questions about abuse, and fear of offending the patient [40]. Sexual health practitioners in HERMES expressed a desire for reinforcement training that would provide opportunities to discuss real-life cases they encountered in their work. This highlights the need for ongoing support and professional development in the healthcare response to DVA in gbMSM, a finding which is ubiquitous in the literature on healthcare responses for women affected by DVA [41].

Despite these barriers, the training had a positive impact on identification practices. The addition of a DVA code in the male patient proforma prompted practitioners to ask about it and the inclusion of GALOP staff in the training increased practitioners' motivation. Their increased confidence and preparedness can also be attributed to their knowledge of GALOP and the reassurance that they did not have to distinguish between 'victim' and 'perpetrator' when there was uncertainty surrounding this. Inadequate response has been linked with certain services, such as the police, being inept at identifying the perpetrator of violence [38].

Practitioners in our study believed that sexual health services were good entry points for gbMSM to seek assistance for DVA which resonates with the views of men in this study. Men

felt that this was preferable to general practice settings where their sexual orientation was never discussed in relation to their health [42]. Practitioners said that they already pose sensitive questions about sexual history and mental wellbeing and found it viable to integrate questions about DVA into their existing practices.

This perspective highlights the potential for sexual health services to provide a comprehensive approach to addressing the unique needs of gbMSM experiencing DVA. However, there were divergent perspectives between practitioners and men regarding whether enquiry for DVA should be conducted routinely or only when patients present with symptoms consistent with experiences of DVA (e.g., repeat sexually transmitted infections, engaging in risky sexual behaviour, depression).

In the evaluation of HERMES, practitioners were in favour of asking all gbMSM about DVA to prevent patients feeling judged. They also considered it a more feasible approach to integrate into their practice rather than relying on particular symptoms as triggers for inquiry which they found difficult to remember and judge. In contrast, the formative phase of our study revealed that only a third (34.7%) of men in the Relationship and Health survey agreed with the statement that sexual health practitioners should ask all patients whether they have been hurt or frightened by a partner, and 62.6% supported selective enquiry based on symptoms [27]. Furthermore, interviews with men attending this service indicated that some were concerned that the clinic was too hectic to ask everyone and respond in a sensitive manner, and it was important to choose the right moment to ask because trust and boundaries in clinical encounters were negotiated over time. In addition, it was felt that on-site support might be needed if asking all patients as some of them may be in need of immediate help [42].

The ADViSE (Assessing for Domestic Violence in Sexual Health Environments) pilot intervention aimed to improve identification and management of DVA in a generic sexual healthcare service in the UK. A prompt to ask a question about DVA was embedded within the clinic's Electronic Patient Record system and received mixed views. Some clinicians or patients felt it was an irritant, whilst others found it to be a helpful tool compelling clinicians to ask the question. They introduced the topic and phrased questions in ways that would prevent causing offence [43]. Similar to our study, clinicians in ADViSE wanted refresher training with feedback on disclosure rates and further discussion. They also identified a need for strong leadership and key points of contact within the healthcare setting to drive the intervention, highlight its importance and deal with operational queries. In HERMES, two clinical co-trainers (a doctor and a senior health advisor) supported implementation by engaging staff in the training and listening to their concerns. However, the role was not formalised and therefore it was unclear what functions they needed to fulfil in order to demonstrate effective leadership. The role of leadership and management in the health system response to violence against women has been identified as one of the key mechanisms for supporting implementation and ensuring acceptance and sustainability [44].

There are a few limitations to our study. HERMES was an uncontrolled before and after pilot study and further testing with more robust designs is needed to estimate the effect of the intervention. The study was conducted in a London-based sexual health service and may not represent the experiences of practitioners and service users in other geographical locations. Although the clinic selected for the study served the broader LGBT population, we chose to focus specifically on the provision of services for gbMSM to respond to the needs of the service. During the inception phase of the study consultations with clinic staff revealed that very few lesbian and transgender patients attended the LGBTQ specialist clinic, which would have made it impossible to obtain an adequate sample size within the study timeline. Additionally, staff members identified DVA among gbMSM as a frequent issue that arose during consultations. They expressed a lack of preparedness to support these patients adequately or to refer

them to appropriate services. This decision may limit the transferability of our results to the experiences of practitioners providing services to other members of the LGBT community.

Qualitative interviews with a sub-sample of gbMSM who completed the Relationships and Health Survey were conducted during the first phase of the study, prior to the implementation of HERMES. Given a more extended timeframe for implementation and evaluation, it would have been possible to interview men who disclosed current experiences of DVA during HERMES. This would have allowed for an exploration of their perspectives on being asked about DVA and their motivations for using or not using the referral service. The PIM survey and semi-structured interviews with healthcare practitioners did not elicit information about their sexual orientation, which may have limited our understanding and insight into how views differed among practitioners with diverse sexual orientations.

Finally, the hospital database did not include information on sexual orientation, which means some of the men attending the AFC and whose medical records we audited to assess increases in DVA among gbMSM might have been heterosexual. For the formative research of this study, we conducted a survey in both regular clinics and a specialised LGBT clinic [27]. All men who participated in the survey from the specialised clinic self-identified as gbMSM, reassuring us that the majority of men attending this clinic self-identify as gbMSM.

Further research is needed regarding how gbMSM would like to receive support when disclosing DVA within a sexual health setting and what support practitioners in those settings need. Some studies highlight the importance of congruence between a practitioner's and a patient's sexual minority status in facilitating the disclosure of sensitive information (xx). However, it is worth noting that a related paper from the formative study, based on interviews with gbMSM from this same clinic, found that men were more comfortable talking to female practitioners [42] furthermore a qualitative review of the literature on help seeking by men who have been exposed to DVA found a consistent preference for receiving help from a female professional across studies and settings [45]. Both studies found that factors such as continuity of care, the interpersonal skills of the practitioner and the attention given to building trust and rapport were considered more important than gender or sexual orientation[42, 45]. Repeating HERMES or conducting future intervention with a cohort of self-identified gbMSM practitioners might help determine if congruence of sexual minority status between practitioner and patient leads to more precise DVA identification and referral.

There is some evidence that apps show promise in engaging gbMSM in sexual health promotion interventions. Given the popularity of geo-social networking apps among gbMSM for meeting sexual partners [46], there may be scope to develop apps that promote healthy relationship behaviour and direct men to specialist DVA services. However, formative research is needed to explore gbMSM perspectives, including the potential risks of escalating abuse if the perpetrator becomes aware that their partner is actively seeking help and barriers to getting support from specialist DVA advocates outside the clinic setting. WHO guidelines on healthcare for women subjected to intimate partner violence or sexual violence advocates a *first line response* denoted by the acronym LIVES [24]. This encompasses listening without judgement (L), inquiring about needs and concerns (I), validating women's experiences whilst reassuring them that they are not to blame (V), enhancing women's safety (E), and offering in-depth case support referrals (S). In the context of sexual healthcare for gbMSM experiencing DVA, this role could be undertaken by skilled-up health advisors who can act as the initial point of referral within the healthcare service and be the link to liaison with external services.

Health advisors already have a role in providing information, advice and counselling to patients diagnosed with a sexually transmitted infection, requiring a non-judgemental approach. The role offers opportunities for building relationships of trust which is critical for discussions about DVA. Much can be learned from the growing evidence base on health

system response to violence against women, which indicates that interventions that focus only on change at the individual provider or clinic level are insufficient for promoting sustainable changes in practice [41, 44, 47]. A comprehensive health system approach is required to provide quality sexual health care for individuals who identify as LGBTQ that experience DVA. Until further research evidence is generated, sexual health services including HIV clinics should, at the very least, display information on LGBTQ DVA and support services.

## Supporting information

**S1 Fig. HERMES flowchart.**
(PDF)

## Acknowledgments

The authors would like to express their gratitude to GALOP, Respect UK, James Rowland and the clinical co-trainers for their valuable contributions to the HERMES intervention. Additional appreciation is extended to the practitioners who generously dedicated their time to participate in the interviews, and the health advisors who conducted the audit of medical records.

## Author Contributions

**Conceptualization:** Gene S. Feder, Loraine J. Bacchus.

**Data curation:** Ana Maria Buller, Alexandra Bleile, Petra J. Brzank, Loraine J. Bacchus.

**Formal analysis:** Ana Maria Buller, Giulia Ferrari, Alexandra Bleile, Petra J. Brzank, Loraine J. Bacchus.

**Funding acquisition:** Gene S. Feder, Loraine J. Bacchus.

**Investigation:** Ana Maria Buller, Loraine J. Bacchus.

**Methodology:** Ana Maria Buller, Giulia Ferrari, Gene S. Feder, Loraine J. Bacchus.

**Project administration:** Gene S. Feder, Loraine J. Bacchus.

**Resources:** Gene S. Feder.

**Supervision:** Ana Maria Buller, Loraine J. Bacchus.

**Validation:** Ana Maria Buller, Loraine J. Bacchus.

**Writing – original draft:** Ana Maria Buller, Loraine J. Bacchus.

**Writing – review & editing:** Ana Maria Buller, Giulia Ferrari, Alexandra Bleile, Gene S. Feder, Petra J. Brzank, Loraine J. Bacchus.

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
