## [Decision Letter · Decision Letter 0]

7 Jun 2024

PONE-D-23-38821HEalth professionals Responding to MEn for Safety (HERMES): Mixed method evaluation of a pilot sexual health intervention for gay, bisexual and other men who have sex with men experiencing domestic violence and abusePLOS ONE

Dear Dr. Buller,

Thank you for submitting your manuscript to PLOS ONE. After careful consideration, we feel that it has merit but does not fully meet PLOS ONE’s publication criteria as it currently stands. Therefore, we invite you to submit a revised version of the manuscript that addresses the points raised during the review process.

We look forward to receiving your revised manuscript.

Kind regards,

Michelle L. Munro-Kramer, PhD, CNM, FNP-BC, FAAN

Academic Editor

PLOS ONE

Additional Editor Comments:

Thank you for your submission to PLOS One. I apologize about the very long review time. We attempted to secure two reviewers for this manuscript, but after months of trying have decided to move forward with one review to prevent any further delays. Please see Reviewer #1 comments for minor revisions before resubmission.

Reviewers' comments:

Reviewer's Responses to Questions

**Comments to the Author**

1. Is the manuscript technically sound, and do the data support the conclusions?

Reviewer #1: Yes

2. Has the statistical analysis been performed appropriately and rigorously? 

Reviewer #1: Yes

3. Have the authors made all data underlying the findings in their manuscript fully available?

Reviewer #1: No

4. Is the manuscript presented in an intelligible fashion and written in standard English?

Reviewer #1: Yes

5. Review Comments to the Author

Reviewer #1: Thank you for the opportunity to review this manuscript. The HERMES intervention is an important development in the DVA space for gbMSM, and will make an important contribution to this literature. However, some concerns regarding the age of this data combined with minor concerns throughout the methods and discussion require revision before this paper is acceptable for publication. I look forward to re-reviewing this work and congratulate the authors on this study.

Introduction

- I am usure why there is a hyphen before gbMSM in line 69?

- Given the paper is specifically about gbMSM, I suggest sticking with a singular acronym (gbMSM) versus including the broader LGBT umbrella (line 86). If the authors choose to continue with a broader term, using a single one (LGBT vs LGBTQ) is recommended (line 108).

- If the clinic in which HERMES was conducted includes the broader LGBT community, why do the authors focus specifically on gbMSM?

- There is a notable lack of data on DVA in gbMSM in the Introduction. While representative estimates remain unavailable, data from previous studies of the prevalence and typologies of violence seen in gbMSM would be helpful.

Methods

- Line 143- more information on what the training entailed would be helpful, as well as whether the training was conducted by gbMSM themselves, specifically covered aspects of DVA in gbMSM that are different to females, and if any of the 31 practitioners self-identified as gbMSM.

- Line 163- the dates indicated in this paragraph are from 2012. Do the authors mean to say HERMES was conducted more than a decade ago? This is a serious limitation and renders the 2014 policy changes given in the Introduction moot.

- Line 191- citations providing substantiation regarding bidirectionality are needed. Suggest using Kirschbaum, A. L., Metheny, N., Skakoon-Sparling, S., Grace, D., Yakubovich, A. R., Cox, J., ... & Hart, T. A. (2023). Syndemic factors and lifetime bidirectional intimate partner violence among gay, bisexual, and other sexual minority men. LGBT health, 10(S1), S89-S97. As a starting point.

- Line 229- it is unclear whether all self-identified males are a) cisgender; and b) gay or bisexual. Is it possible many of these men were cisgender and heterosexual?

- It is unclear from the Methods whether those who completed the training were also the staff responsible for assessing DVA and referring patients to GALOP. If the authors can provide more context on the processes of the clinic in which the pilot was conducted, this would help the readers understand if the trained providers were actually the ones putting this into practice

Results

- I suggest adding the first paragraph of the Results section to the Methods section instead, given that this paragraph details the “mixing” of the authors’ methods rather than the results of said analyses.

- Line 363- providing the specific finding (value, p-value) of the Wilcoxon test would be helpful

- Line 386: column headers for Table 4 are missing

- A striking finding is that practitioners did not increase their assessment of DVA among males post-training, but did increase their assessment of females, more information on what the training included could help illuminate this.

- The quote beginning on line 447 is especially interesting and provides insight into potential future directions for this research.

Discussion/Conclusion

- The marginal increase in case identification may be due to factors outside the investigators’ control, and could stem from the mixture of trained and untrained practitioners present in the clinic. It is likely that many patients of this clinic know each other and speak to one another about the sensitization of staff to the issue of DVA. Mixed approaches to enquiry and referral may lead to an underestimation of the true effect of the HERMES intervention

- It seems the congruence between practitioner and patient re: sexual minority status is underappreciated in this section. gbMSM (just like any other minority) feel more comfortable disclosing sensitive information to members of their own community, which leads to more precise care and referral. Understanding who these practitioners were is critical to assessing the success of HERMES. Given that the majority of practitioners were female suggests many patients may have felt uncomfortable disclosing and being referred regardless of the practitioner’s training. Repeating HERMES with a cohort of self-identified gbMSM practitioners may be one recommendation of this study.

6. PLOS authors have the option to publish the peer review history of their article (what does this mean?). If published, this will include your full peer review and any attached files.

Reviewer #1: **Yes: **Nicholas Metheny

---

## [Author Response · Author response to Decision Letter 0]

21 Aug 2024

Reviewer #1: 

Thank you for the opportunity to review this manuscript. The HERMES intervention is an important development in the DVA space for gbMSM, and will make an important contribution to this literature. However, some concerns regarding the age of this data combined with minor concerns throughout the methods and discussion require revision before this paper is acceptable for publication. I look forward to re-reviewing this work and congratulate the authors on this study.

Thank you very much for your positive review, we look forward to addressing your comments which will strengthen the paper.

Introduction 

1. I am unsure why there is a hyphen before gbMSM in line 69?

We have removed it.

2. Given the paper is specifically about gbMSM, I suggest sticking with a singular acronym (gbMSM) versus including the broader LGBT umbrella (line 86). If the authors choose to continue with a broader term, using a single one (LGBT vs LGBTQ) is recommended (line 108).

Thank you for this suggestion. We have used the term gbMSM throughout.

3. If the clinic in which HERMES was conducted includes the broader LGBT community, why do the authors focus specifically on gbMSM?

During initial consultations with the sexual health service, the research team were advised that very few lesbians and transgender patients use the LGBTQ specialist clinic (i.e. it would have taken years to get even a small sample). Moreover, clinicians reported that DVA among gbMSM was an issue that arose during consultations, or at times they suspected DVA. However, they did not have the training, clinical guidance or knowledge of referral pathways to support them. For these reasons, the study focussed on experiences of DVA among gbMSM. We have added the following text in the discussion (lines 750-759 ) to acknowledge this as a limitation:

“Although the clinic selected for the study served the broader LGBT population, we chose to focus specifically on the provision of services for gbMSM to respond to the needs of the service. During the inception phase of the study consultations with clinic staff revealed that very few lesbian and transgender patients attended the LGBTQ specialist clinic, which would have made it impossible to obtain an adequate sample size within the study timeline. Additionally, staff members identified DVA among gbMSM as a frequent issue that arose during consultations. They expressed a lack of preparedness to support these patients adequately or to refer them to appropriate services. This decision may limit the transferability of our results to the experiences of practitioners providing services to other members of the LGBT community.”

4. There is a notable lack of data on DVA in gbMSM in the Introduction. While representative estimates remain unavailable, data from previous studies of the prevalence and typologies of violence seen in gbMSM would be helpful.

Thank you for this suggestion, we have included information from two systematic review and meta-analysis reporting on victimisation and perpetration of all form of IPV among gbMSM across different recall periods (lines 73 to 79)

“A systematic review and meta-analysis conducted by Buller and colleagues in 2013 {Buller, 2014 #99} reported a pooled lifetime prevalence of intimate partner violence (IPV) among gbMSM at 48% (95% CI: 31.23%-64.99%), encompassing physical, psychological, and/or sexual violence. More recently, a meta-analysis by Liu and colleagues {Liu, 2021 #295} found a pooled prevalence of 33% (95% CI: 28%-39%) for IPV victimization, including all types of IPV and various recall periods. Liu and colleagues also reported a pooled prevalence of IPV perpetration among gbMSM at 29% (95% CI: 17%-40%) {Liu, 2021 #295}.”

Methods

5. Line 143- more information on what the training entailed would be helpful, as well as whether the training was conducted by gbMSM themselves, specifically covered aspects of DVA in gbMSM that are different to females, and if any of the 31 practitioners self-identified as gbMSM 

Although we know that some trained staff identified as gbMSM, the pre-post PIM survey did not ask health practitioners whether or not they identified as gbMSM. On reflection, this is a limitation in the analysis and we have included this in the discussion (lines 765 to 768) by including the following text:

“The PIM survey and semi-structured interviews with healthcare practitioners did not elicit information about their sexual orientation, which may have limited our understanding and insight into how views differed among practitioners with diverse sexual orientations”. 

Re the training content, the training included a group discussion about what is different for, or specific to gbMSM in terms of their experiences of domestic violence, and also what is different for, or specific to men who don’t identify as either, but do have sex with men. We have included more information on the training content (lines 137 to 151) summarised as follows:

The HERMES training for sexual health practitioners included content on: 

i) how to identify signs and symptoms consistent with DVA in gbMSM; 

ii) group discussion about what is different for, or specific to gbMSM experiences of DVA compared with heterosexual women and men who experience DVA, and what is different for, or specific to men who do not identify as gay or bisexual, but do have sex with men; 

iii) national prevalence of DVA in gbMSM, health impacts and risk factors for DVA; 

iv) data on the prevalence of DVA among gbMSM attending the sexual health service based on a survey conducted in Phase 1 of the study, and their views on health practitioners asking gbMSM about DVA; 

v) practice in asking questions about DVA using a tailored designed flowchart (Supplementary file 1); 

vi) documentation of DVA in the patient clinic proforma including whether the patient was asked about DVA, their response and whether a referral was offered;

vii) and a care pathway which included referral to the clinic health advisors who would act as the link to liaison with GALOP, a London-based anti-violence and abuse charity for lesbian, gay, bisexual and transgender (LGBT) people and finally 

viii) ongoing support for health care practitioners from a lead health advisor and a doctor.

6. Line 163- the dates indicated in this paragraph are from 2012. Do the authors mean to say HERMES was conducted more than a decade ago? This is a serious limitation and renders the 2014 policy changes given in the Introduction moot.

Thank you for this comment. On reflection we don’t think the text on the Home Office Strategy on Violence Against Women and Girls Home Office is relevant and we have removed it. However, we would like to retain the text on the NICE guidelines that precede this text. The extent to which the UK NICE guidelines have been implemented, as they pertain to gbMSM, is unknown and we have mentioned this in the text. Furthermore, we consider that the fact that this study is over a decade old does not detract from its significance in the current research and policy context. We have amended the text (lines 104 to 113) to clarify that there has been a notable lack of research on healthcare interventions that focus on gbMSM living with DVA since the study reported in this paper, which was conducted between 2009 and 2011. This has resulted in the experiences of gbMSM being overlooked in current global discussions about how to strengthen the healthcare response to patients who experience DVA. In our experience to date, the health system response to DVA is firmly focussed on heterosexual women as victims.

“There has been a growing body of evidence over the last two decades exploring how healthcare services can best respond to DVA experienced by women in heterosexual relationships. However, there has been a notable lack of research on healthcare interventions that focus on the needs of gbMSM living with DVA, since the study reported in this paper, which was conducted between 2009 and 2012. This has resulted in the experiences of gbMSM being overlooked in current global discussions about how to strengthen the healthcare response to patients who experience DVA. The extent to which the UK NICE guidelines have been implemented in UK healthcare settings, as they pertain to gbMSM, is unknown. Consequently, there remains a critical gap in the current evidence base and further research is urgently needed to address this disparity.” 

7. Line 191- citations providing substantiation regarding bidirectionality are needed. Suggest using Kirschbaum, A. L., Metheny, N., Skakoon-Sparling, S., Grace, D., Yakubovich, A. R., Cox, J., ... & Hart, T. A. (2023). Syndemic factors and lifetime bidirectional intimate partner violence among gay, bisexual, and other sexual minority men. LGBT health, 10(S1), S89-S97. As a starting point.

Thank you very much for raising this point. We have now included the suggested citation and added an additional citation of a systematic review on the issue by Messinger 2015 (line 235)

“bidirectional violence in this population {Kirschbaum, 2023 #294;Messinger, 2018 #293}.”

8. Line 229- it is unclear whether all self-identified males are a) cisgender; and b) gay or bisexual. Is it possible many of these men were cisgender and heterosexual?

Thank you for raising this point. The database did not include information on the sexual orientation or gender identity of participants. Use of the specialised LGBT clinic was reserved for gbMSM, lesbian and transgender people because the clinic only ran every Tuesday evening (hence the name ‘After Five Clinic’). Use of the clinic by cisgender/heterosexual males was discouraged by staff because there were two generic sexual health clinics in the hospital that were open five days a week. For the broader study we conducted a survey in the generic and specialised clinic and 100% of men who agreed to participate in the survey from the specialised clinic self-identified as gbMSM. Therefore, we are confident that most men attending this clinic self-identified as gbMSM. Whilst there may be a possibility that some of these men were cisgender and heterosexual (as a way of getting an appointment quickly), we consider that this was rare because it was discouraged by staff. We have included in limitations as follows (lines 769 to 774)

“Finally, the hospital database did not include information on sexual orientation, which means some of the men attending the AFC and whose medical records we audited to assess increases in DVA among gbMSM might have been heterosexual. For the formative research of this study, we conducted a survey in both regular clinics and a specialised LGBT clinic {Bacchus, 2017 #274}. All men who participated in the survey from the specialised clinic self-identified as gbMSM, reassuring us that the majority of men attending this clinic self-identify as gbMSM.”

9. It is unclear from the Methods whether those who completed the training were also the staff responsible for assessing DVA and referring patients to GALOP. If the authors can provide more context on the processes of the clinic in which the pilot was conducted, this would help the readers understand if the trained providers were actually the ones putting this into practice.

Thank you for this comment we have now added the following paragraphs to the ‘Setting’ section in Methods (lines 173 to 179):

“Before HERMES, the process at the After Five Clinic involved referring suspected cases of DVA to one of the Health Advisors, with these cases being manually recorded in the medical records without a specific place for DVA documentation, meaning it could be noted anywhere in the records. During HERMES, we maintained the basic process of referring suspected or disclosed DVA cases to a Health Advisor (as described in the previous section on training). The Health Advisor would provide a first-line response by discussing the patient’s situation further and offering to make a referral to GALOP.

Only trained healthcare practitioners were required to implement HERMES in the specialist LGBTQ sexual health clinic. These trained practitioners were not required to implement HERMES in the generic sexual health clinics, which were primarily used by heterosexual patients, although we know that gbMSM would occasionally use those clinics if they were unable to get a timely appointment at the LGBTQ clinic. Staff were sometimes required to rotate between the generic and LGBTQ clinics, but those who did not receive training were not required to participate in HERMES.”

Results

10. I suggest adding the first paragraph of the Results section to the Methods section instead, given that this paragraph details the “mixing” of the authors’ methods rather than the results of said analyses.

Thank you for this suggestion. We have split this paragraph in two and moved the second part to methods as it focusses on the mixed methods analysis and its justification. We have however kept the first part of the paragraph in the results section as it describes the structure of the subsequent sections.

11. Line 363- providing the specific finding (value, p-value) of the Wilcoxon test would be helpful

Thank you for this suggestion, we have amended the text in lines 409 to 420 as follows:

“With regards to changes in practitioners’ preparedness (i.e. asking, responding, identifying and referring) to deal with DVA, a non-parametric Wilcoxon signed-rank test suggested increased self-reported ability in dealing with all patients that may be at risk of involvement in DVA episodes in the PIM survey (Table 4). Whilst there was an increase in perceived preparedness across all patient groups, the largest improvement was reported for gbMSM patients (p-values ranging from 0.0002 to 0.0064 across the four components of preparedness, see Table 4). The smallest improvement was observed in the management of female patients, in particular we did not observe any improvement in practitioners’ preparedness to ask for female patients about their experiences of victimisation (p=0.1458).”

12. Line 386: column headers for Table 4 are missing

Thank you for spotting that, we have added the headings and a footnote to the table. 

13. A striking finding is that practitioners did not increase their assessment of DVA among males post-training, but did increase their assessment of females, more information on what the training included could help illuminate this.

Thank you for highlighting this result. In summary, we observed:

a) Improved preparedness across the board of patient groups, but the change was more modest for female patients and greatest for gbMSM patients in line with what you would expect for an intervention that focussed on gbMSM, and sexual health providers that had already received training on women DVA. 

b) Conversely, and as the reviewer points out we found that for identification (asking about experience of DVA who showed sigs of DVA) was either marginal or non-existent for men including gbMSM.

We hope that the modifications made to the text in response to reviewer’s point 10 as well as the inclusion of the following text in the discussion 648 to 655 help clarify these points: 

“The mixed method evaluation of HERMES demonstrated that it was feasible and acceptable to staff. The training increased sexual health practitioners’ confidence in asking all patients groups, including gbMSM, about DVA and responding to disclosures, with the greatest impact on their preparedness to handle gbMSM cases. The smallest improvement was observed in the management of female patients, with no improvement in the practitioners’ ability to ask about female experiences of victimisation. This lack of improvement can be explained by the fact that sexual health practitioners had previously received training in a separate pilot intervention for female patients (2005-2007) called MOZAIC Women’s Wellbeing Project (33).”

Lines 658 to 664:

“However, both the PIM survey and the pre- and post-intervention audit of medical records showed none or only a marginal increase in identified cases of DVA among men including gbMSM. Conversely, the PIM survey showed that identification of female patients increased after HERMES. Identification requires a higher level of skills an

---

## [Decision Letter · Decision Letter 1]

15 Oct 2024

HEalth professionals Responding to MEn for Safety (HERMES): Mixed method evaluation of a pilot sexual health intervention for gay, bisexual and other men who have sex with men experiencing domestic violence and abuse

PONE-D-23-38821R1

Dear Dr. Buller,

We’re pleased to inform you that your manuscript has been judged scientifically suitable for publication and will be formally accepted for publication once it meets all outstanding technical requirements.

Kind regards,

Michelle L. Munro-Kramer, PhD, CNM, FNP-BC, FAAN

Academic Editor

PLOS ONE

Additional Editor Comments (optional):

Thank you for your patience. We are happy to accept this manuscript for publication.

Reviewers' comments:

Reviewer's Responses to Questions

**Comments to the Author**

1. If the authors have adequately addressed your comments raised in a previous round of review and you feel that this manuscript is now acceptable for publication, you may indicate that here to bypass the “Comments to the Author” section, enter your conflict of interest statement in the “Confidential to Editor” section, and submit your "Accept" recommendation.

Reviewer #1: (No Response)

2. Is the manuscript technically sound, and do the data support the conclusions?

Reviewer #1: Yes

3. Has the statistical analysis been performed appropriately and rigorously? 

Reviewer #1: Yes

4. Have the authors made all data underlying the findings in their manuscript fully available?

Reviewer #1: No

5. Is the manuscript presented in an intelligible fashion and written in standard English?

Reviewer #1: Yes

6. Review Comments to the Author

Reviewer #1: The authors have addressed my comments and I now feel the manuscript is ready for publication. Should the authors choose to include references to literature that shows LGB patients are often more comfortable disclosing sensitive information to LGB providers (as requested in one comment), some potential citations are below. Acceptance of the manuscript is not incumbent on including these or further addressing the comment in question, however.

Adams J, McCreanor T, Braun V. Doctoring New Zealand’s gay men. N Z Med J

2008; 121(1287): 11–20

Sharek DB, McCann E, Sheerin F, et al. Older LGBT people’s experiences and

concerns with healthcare professionals and services in Ireland. Int J Older

People Nurs 2015; 10(3): 230–240. [Epub 2014].

Brooks, H., Llewellyn, C. D., Nadarzynski, T., Pelloso, F. C., Guilherme, F. D. S., Pollard, A., & Jones, C. J. (2018). Sexual orientation disclosure in health care: a systematic review. British Journal of General Practice, 68(668), e187-e196.

7. PLOS authors have the option to publish the peer review history of their article (what does this mean?). If published, this will include your full peer review and any attached files.

Reviewer #1: No

---

## [Editor Report · Acceptance letter]

4 Nov 2024

PONE-D-23-38821R1 

PLOS ONE

Dear Dr. Buller, 

I'm pleased to inform you that your manuscript has been deemed suitable for publication in PLOS ONE. Congratulations! Your manuscript is now being handed over to our production team.

Kind regards, 

on behalf of

Dr. Michelle L. Munro-Kramer 

Academic Editor

PLOS ONE